# A Review of Food Texture Modification among Individuals with Cerebral Palsy: The Challenges among Cerebral Palsy Families

**DOI:** 10.3390/nu14245241

**Published:** 2022-12-09

**Authors:** Sakinah Kamal, Sazlina Kamaralzaman, Shobha Sharma, Nurul Hazirah Jaafar, Phei Ming Chern, Nurul Izzaty Hassan, Hasnah Toran, Noor Akmal Shareela Ismail, Ghazali Yusri, Nur Hana Hamzaid

**Affiliations:** 1Center for Rehabilitation and Special Needs Studies (iCaRehab), Faculty of Health Sciences, Universiti Kebangsaan Malaysia (UKM), Jalan Raja Muda Abdul Aziz, Kuala Lumpur 50300, Malaysia; 2Dietetics Program, Faculty of Health Sciences, Universiti Kebangsaan Malaysia (UKM), Jalan Raja Muda Abdul Aziz, Kuala Lumpur 50300, Malaysia; 3Center for Healthy Aging and Wellness (H-Care), Faculty of Health Sciences, Universiti Kebangsaan Malaysia (UKM), Jalan Raja Muda Abdul Aziz, Kuala Lumpur 50300, Malaysia; 4Department of Nutrition Sciences, Kulliyyah of Allied Health Sciences, International Islamic University Malaysia (IIUM), Jalan Sultan Ahmad Shah, Bandar Indera Mahkota, Kuantan 25200, Malaysia; 5Department of Rehabilitation Medicine (Paediatric Rehabilitation), Hospital Rehabilitasi Cheras, Jalan Yaacob Latif, Bandar Tun Razak, Cheras, Kuala Lumpur 56000, Malaysia; 6Department of Chemical Sciences, Faculty of Science & Technology, Universiti Kebangsaan Malaysia (UKM), Bandar Baru Bangi 43600, Malaysia; 7Faculty of Education, Universiti Kebangsaan Malaysia (UKM), Bandar Baru Bangi 43600, Malaysia; 8Department of Biochemistry, Faculty of Medicine, Universiti Kebangsaan Malaysia (UKM), Jalan Yaacob Latif, Bandar Tun Razak, Cheras, Kuala Lumpur 56000, Malaysia; 9Akademi Pengajian Bahasa, Universiti Teknologi Mara (UiTM), Shah Alam 40450, Malaysia; 10Malaysian Advocates for Cerebral Palsy (MyCP), No 4., USJ3/4X, USJ3, Subang Jaya 47600, Malaysia

**Keywords:** cerebral palsy, swallowing difficulties, dysphagia, food texture modification

## Abstract

Individuals with cerebral palsy (CP) frequently present with multiple feeding problems, which may require food texture modification to ensure safe feeding. This review aims to explore the challenges individuals with CP and their caregiver’s face and recommend modified food textures to ensure safety and improve the quality of life and nutritional status. A systematic search was carried out through four databases (i.e., EBSCO (Medline), PubMed, Science Direct, and Web of Science) between January 2011 and May 2022. Out of 86 articles retrieved, seven were selected based on keywords and seven other studies through manual search-five cross-sectional studies, two qualitative studies, one correlational study, one mixed method study, one case-control study, two sections of books, and two educational materials. The findings suggest that preparation and intake of food with modified texture play a necessary role in the safety of swallowing in addition to physical, social, and environmental aspects. Safety was found to be the crucial part of the food texture modification provision besides considering the stress of the caregivers and the nutritional status of individuals with CP. Currently, there are no standard guidelines available pertaining to food texture modification. This led to uncertainties in the dietary provision among caregivers, which may lead to undernourishment. Hence, standard guidelines relating to food texture modification that focuses on food preparation and menus with calorie and nutrient information are timely to be developed.

## 1. Introduction

Cerebral palsy (CP) is defined as permanent but non-progressive disorders, traumas, or abnormalities in the developing/immature brain that lead to activity limitation [1]. According to data from European countries, the average incidence of CP is 2.08 per 1000 births, with the risk of CP increasing with early delivery and low birth weight [2]. Other risk factors include trauma during birth, aided using forceps or vacuum, and complications after birth, such as hyperbilirubinemia and hypoglycemia [3]. Note that CP is classified based on neurological syndromes, the location of the insult, and the severity of symptoms. The forms of CP include diplegia, hemiplegia, bilateral hemiplegia (tetraplegia), ataxia, dyskinesia, and mixed types. The Surveillance of Cerebral Palsy in Europe (SCPE) recommends the classification of spasticity involving both lower and upper limbs using the five levels of the Gross Motor Function Classification System (GMFCS). It was first discovered by Palisano et al. [4], where the higher levels of GMFCS indicate a higher dependency on the caregivers [5], including that for feeding.

Dysphagia is a feeding problem prevalent in the CP population. It is caused by oral motor dysfunction, abnormal neurological maturity, and gastrointestinal problems. A meta-analysis revealed that the pooled prevalence of swallowing was 50.4%, and feeding problems were 53.3% among individuals with CP [6]. Hence, researchers suggested early detection of dysphagia in CP children as it may reduce the risk of health problems, optimize nutritional intake, and facilitate rehabilitation treatment. A simple, proactive, and rapid tool for the early identification of dysphagia is a validated questionnaire that can be used to screen eating difficulties [7]. An instrumental procedure with barium solution while swallowing food and drink under X-rays has been considered a gold standard in assessing dysphagia among patients. This applies to the children population with a possibility of dysphagia to also be examined using a procedure such as videofluoroscopy (VFSS) in which barium is being used to gather information about the pharyngeal phase of the swallow and potential silent aspiration. Dysphagia detection using multidisciplinary approaches has been good practice as it looks at multiple aspects that lead to a wholesome treatment [8,9,10]. These include an in-depth physical and clinical examination by a neurologist on the medical and social history, type of medication, and current respiratory problems, in addition to further investigation by the speech therapist focusing on oral motor processes and the dietitian’s assessment of the nutrition and hydration status. Subsequently, the dysphagia assessment via online rehabilitation merged with current advanced technology [11] with adequate resources and infrastructures [12]. Upon formal diagnosis, of utmost importance would be to choose treatment or intervention to improve swallowing safety while maintaining nutrition.

Although some individuals with dysphagia can still eat orally, food texture modifications are frequently recommended to prevent complications arising from the aspiration of foods and fluids into the lung. It is important to note that modification of food textures not only complicates meal preparation and results in lengthy feeding times but can compromise the nutritional status of children with CP if prescribed incorrectly [7]. Food example, thick or lumpy food can cause residue in the pharynx when pharyngeal motility is decreased. This has the potential to cause aspiration. Similar issues can arise with the type and size of the bolus presented [13]. The role of family members in dysphagia management is crucial, and they need to be involved in determining goals and activities in a feeding plan to ensure the plan is achievable. Sousa et al. reported that one area of concern was the caregivers’ lack of standards during meal preparation which led to an unbalanced diet [14]. On the other hand, a qualitative study conducted by Marques et al. [15] listed out challenges experienced by parents in the feeding process, which pointed to the main issues of food spilling, choking, and vomiting that leads to parents’ anxiety. Similarly, an unbalanced diet, food monotony, shame, challenging access to tailored outside food, inadequate technical aids, time, cooking food separately, costs and lack of social support were among the difficulties faced by parents of the children [15,16]. Parents tend to prepare food with the same range that is faster to cook as they need to cook separately, which leads to less variety of food/nutrients [15]. Another research found that a longer feeding time contributed to inadequate calorie intake [17].

Modifications to food consistency are one plausible management for individuals with feeding problems. According to Benfer [18] and colleagues, 39% of 99 young children were on modified food textures among individuals with CP. Therefore, several nutrition modules have been developed to assist children with eating and swallowing difficulties. The Integrated Nutrition Training Program (INTP), which was developed in Malaysia, was curated for caregivers/trainers of special needs children, including CP. Consisting of five components, namely nutrition education, eating equipment, nutritional stimulation, food texture modification, as well as mealtime interactions and behaviors. The research revealed an increase in caregiver knowledge after implementing the program [19]. Prior research by Chen et al. [20] also designed the Intervention Program (IP) among 45 trainers in a rehabilitation center by evaluating their knowledge, behaviors, and attitudes for a specified nutrition module. The content of the nutrition module is presented in the form of innovative demonstrations and learning techniques, including healthy eating, optimal weight, and nutrition for children with special needs. The Focus on Early Eating, Drinking and Swallowing (FEEDS) toolkit has been recommended by Parr [21] and colleagues as an intervention for neurologically impaired children with eating, drinking, and swallowing difficulties. Caregivers and health professionals found it advantageous due to components such as adjustment of food or drink characteristics, modification of food or beverage consistency, visual assistance, children’s feeding cue response, and exposure to novel foods and textures. There is another IP involving 20 caregivers at Alexandria Hospital, Egypt, which demonstrated that early exposure to information regarding eating skills, how to feed, communication while feeding, behavior at mealtimes, type of food, the orientation of eating problems, and eating skills (chewing and swallowing) which provides more significant benefits as compared to control groups [22].

The more comprehensive and global framework, the International Dysphagia Diet Standardisation Initiative (IDDSI), was developed in 2013 to standardize the terminology used for food texture modification and liquid consistency. This framework aims to be used in various healthcare settings, cultures, and in different ranges of ages. It contains eight levels (0–7) marked by numbers, text labels and color-coded. Levels 0 to 2 were intended for liquid only with different consistency (0 = thin, 1 = slightly thick, 2 = mildly thick). Meanwhile, levels 3 and 4 share the same characteristics in the form of liquid consistency and food texture. Level 3 in liquid consistency refers to moderately thick, and level 4 refers to extremely thick. Level 3 food texture refers to liquidized, and level 4 refers to pureed. Levels 5 to 7 are transitional in food texture, whereby level 5 is minced and moist, level 6 is soft and bite-sized, and level 7 is easy to chew and regular. A testing method must be used to determine the code and level of IDDSI. For liquid levels 0–2, the IDDSI flow test uses a syringe and measures flow at a stipulated time. For level 3 onwards, additional tests can be done to confirm the texture and consistency with a fork drip test, spoon tilt test, chopstick test or finger test. Note that each level had a clear description of the sizes of the food and characteristics after each testing method was done. Shaw et al. [23] conducted a study to test the feasibility of materials provided by the IDDSI framework. The study was done among 59 university nutrition students who were required to prepare food at Level 4 Pureed Diet and Level 2 Mildly Thick Liquids of IDDSI and compared with other commercial food products with the same characteristics as food prepared by sensory evaluation. Apart from that, four different menus of school meals for children were developed for food. It was found that most of the participants agreed (92%) that the materials and instructions to do food modification and liquid consistency were easy to follow. Furthermore, more than half of the subjects preferred food prepared over commercial products. The study also highlighted that the practice of IDDSI can ensure compliance with the meal by emphasizing safety issues while taking the food with modification [23].

In most nutrition modules that included a food texture modification component, only limited food textures and characteristics such as chopped/minced diet, soft diet, and puree/blended diet were presented. The term “food texture modification” was also utilized interchangeably with “food or liquid consistencies”, referring to altering or modifying characteristics or texture of typical individuals consumed to suit the swallowing difficulties that individuals with CP experienced, thus may have an impact on the correct preparation of modified food and fluid textures by parents and caregivers. The impact of the challenges faced by parents in preparing tailored consistencies remains unclear. This review aims to explore the challenges faced by individuals with CP and their caregivers and the recommendations for modified food textures. A preliminary search was conducted using PubMed, the Cochrane Database of Systematic Reviews, and the Joana Briggs Institute (JBI). Evidence Synthesis was conducted. No ongoing and current systematic or scoping reviews were identified using the related keywords. Thus, this scoping review intends to:Explore the challenges of individuals with CP and their caregivers when preparing modified textured food.Explore the components of nutrition management emphasizing food texture modification.

## 2. Materials and Methods

This scoping review followed Arksey and O’Malley’s foundational methodology for scoping reviews [24]. Levac, Colquhoun, and O’Brien indicated that the search, screen, and report scoping review approach would be adjusted and implemented [25]. Consequently, a PRISMA-P checklist was used to draft a scoping review procedure [26]. The Preferred Reporting Items for Systematic Reviews and Meta-Analyses extension for Scoping Review (PRISMA-ScR) recommendations were followed when reporting the scoping review. The scoping review framework included six stages: (1) identifying the research question; (2) identifying relevant studies; (3) selecting studies to be included; (4) charting information and data from the studies; (5) collecting, summarizing, and reporting the results; and (6) consulting stakeholders and experts [27].

### 2.1. Identifying Research Questions

In the initial phase of the review, research questions were developed using the Population, Concept, and Context (PCC) concept. The PCC table, as presented in Table 1 guided the search for relevant studies and the eligibility criteria. Based on the PCC table, subheadings and synonyms were developed for keyword search in the database using an online thesaurus dictionary (Table 2). For medical terms, the keywords were searched and compared with MeSH terms from the Cochrane database (Appendix A). Search strings with Boolean “AND” and “OR” were used to find the relevant studies and by combining the concept of the keywords.

Research questions for this scoping review are stated as the following:What are the challenges of individuals with cerebral palsy (CP) and their caregivers when preparing modified textured food?What are the components of nutrition management in the recommendation of food texture modification?

### 2.2. Identifying Relevant Studies

The search was limited to January 2011 until May 2022. A longer time frame was selected to cover the topics since not many studies focus on food texture modification among individuals with CP [28]. Other than that, the electronic searches included databases from EBSCOhost (Medline), PubMed, Science Direct, and Web of Science. This scoping review considered experimental, descriptive observational study, and qualitative study designs for inclusion. Relevant literature databases such as Grey Literature Report, OpenGrey, government documents, academic theses/dissertations, a chapter from books, reports, text, opinion papers, and conference abstracts relevant to this review are also included. Subsequently, manual searching was also included pertaining to food texture modification in the population with CP.

The search was further filtered for English language only, human subjects, and full-text availability found in relevant peer-reviewed journals. The articles that do not have access were further forwarded to a university librarian to help search for the paper’s full text. Note that the search strategy, including all identified keywords and index terms, was used in each database and/or information source. A table of search strings is presented in (Appendix B).

The reference list of all included sources of evidence will be screened for additional studies. For the database that has limited use of the Boolean operator, keywords will be selected accordingly. Any review articles and all other secondary sources were excluded from the study. Only original articles with the full manuscript will be included in this review. The search results were downloaded into End-Note software version 20, and duplicated studies were removed.

### 2.3. Selection of Studies

Similarly, studies that included individuals with CP and those who eat textured-modified meals every day were included. Individuals with CP who depended solely on tube feeding, transitional feeding, or combined tube feeding were excluded.

Article screening and selection involved three processes. First, SK screened the title, and SK and NHH screened the abstract/summary and complete text for inclusion criteria. Note that, additional reviewers discussed any discrepancies to ensure that the conclusion was consistent with the scope of this review. The search results and the study inclusion process will be reported in full in the final scoping review and presented in a PRISMA-ScR flow diagram.

### 2.4. Charting Information and Data from the Studies

At this stage, all selected data was extracted from the journal databases and transferred to Microsoft Word to the data extraction table according to year, author (s), country, study location, method (study design, sample size, and instruments used), and findings or key points from the study.

### 2.5. Collating, Summarizing, and Reporting the Results

During data extraction from each source, the initial data extraction table was changed and revised. Apart from that, disagreements between reviewers are resolved through discussion with additional reviewers, and authors of the studies were contacted to obtain missing or additional data. Challenges faced by individuals and families with cerebral palsy are presented in the following table (Table 3).

## 3. Results

### 3.1. Characteristics of Selection Studies

The initial literature screening through various databases revealed that 86 articles matched the keywords. Further screening revealed 14 duplicate records (Figure 1) were removed. From the manual search, seven additional related studies were added. In the consequent screening stage, 52 records were excluded as they did not comply with the topic and inclusion criteria. From the remaining 27 full articles, 14 articles were selected. Similarly, 13 articles were rejected as they did not focus on food texture (n = 1), involved technology application (n = 1), presented Standard Operation Procedure (SOP) for dietitians (n = 1), were unrelated review papers (n = 7), only explained nutritional status (n = 1). The final article did not explain the intervention for dysphagia (n = 2). Therefore, this review yielded ten original articles, five cross-sectional studies, two qualitative studies, one correlational study, one mixed-method study, and one case-control study. Two articles were from a book chapter, and the other two were from educational materials. Note that the studies included were from 10 different countries, with the majority from the USA and Australia (n = 3) and each from Canada, Egypt, Korea, Mexico, Portugal, Spain, The Netherlands, and Turkey. A narrative descriptive approach was used to synthesize the results.

Three studies included in this review were from a cross-sectional study that compared two groups which were the category of individuals with cerebral palsy (CP) (n = 297), and the other category was healthy individuals (n = 192) [35,36,41]. Two cross-sectional studies were done descriptively with no comparison with other groups, including 143 individuals with CP [18,31]. Besides, all the subjects took oral diets. Nutritional status assessments were assessed by anthropometry data which consisted of weight, height, 24-h diet recall with amount and types of food texture taken, and 3-day weighed food record. Among the questionnaires instrument that was used to assess the feeding problem were Dysphagia Disorder Survey (DDS) [42], Schedule Oral Motor Impairment (SOMA) [43], Pre-Speech Assessment Scale (PSAS) [44], and Thomas-Stonell and Greenberg Saliva Severity Scale [45]. For parent’s/caregivers’ self-reported instruments, the Queensland Feeding Questionnaire (CFQ) and Behavioral Pediatric Feeding Scale (BPFAS) were administered. Other than that, a diagnostic instrument using videofluoroscopy (VFSS) was used in a case-control study among individuals with CP to confirm dysphagia [30]. One study conducted a correlational analysis to explore the relationship between feeding difficulties and Gross Motor Function Classification System (GMFCS) classification [32]. Similarly, another study was carried out using a mixed-method exploratory design to examine the challenges of feeding problems faced by caregivers [15].

### 3.2. Challenges of Feeding Problems among Individuals and Families with CP

Caregivers reported facing significant feeding problems with CP, such as choking or gagging during mealtimes, longer feeding time, and incidence of spitting food out from the mouth with more swallowing impairments in the more severe CP [15,18,30,31,32,35,36,41].

A qualitative study of in-depth interviews via telephone among 11 caregivers of children with CP pointed out four themes: 1. A child-centered world in which the child is the focus as the caregivers worry about choking, swallowing, safety, body weight, food preparation and modification, positioning while feeding, and longer feeding time 2. Making decisions that involved problem-solving related to the child’s health status and the need for them to seek knowledge to manage feeding problems. 3. Knowing their child with unique feeding feeds affects the food preparation process, and 4. Seeking and receiving support from family, healthcare professionals, and financial and social support were crucial in handling the feeding problem of their children [29].

Another qualitative study of ten adolescents and young adults with cerebral palsy participants revealed four themes [33]. The mean age was 18.5 years (range 15–23 years old) with participants with GMFCS I (n = 1), GMFCS II (n = 3), GMFCS III (n = 1), GMFCS IV (n = 2) and GMFCS V (n = 3). Three were unemployed, five were students, and the remaining were in special education. The themes that emerged from the analysis were how they perceived eating and drinking difficulties which further summarized the problem and decreased muscle strength that caused difficulties swallowing food with varying textures. Choking while eating is also frequently reported, and the feeding process takes longer, although it requires special utensils to assist the mealtime. The second theme in this study was the challenges when socializing in the community, such as when they eat out, issues regarding accessibility, the environment that needs to be prepared before taking their meals, and old-fashioned utensils used. Other than that, dealing with eating and drinking difficulties appeared to be the third theme, as they tend to avoid food that is difficult to handle when it comes to mealtime. The final theme was the feeling of fear and worry. They felt fear when they experienced symptoms such as coughing and panic when choking. They are also concerned with the treatment that healthcare institutions can offer. Feeding efficiency that is complicated by the process of swallowing difficulties, behavior issues and other physiologic issues will affect total calorie intake and nutritional needs [39,40].

### 3.3. Nutrition Intervention Focusing on Food Texture Modification

A previous study suggested that food texture or energy density modifications might help children with CP and oral impairments [15,18,35,36,37,38,40].

Source from grey literature which included chapters of books and educational materials highlighted several key points in managing challenges with feeding problems among individuals with CP. Neurological individuals with CP might have sensory perception issues and difficulty differentiating the texture and taste of foods when chewing. Hence, texture differences can be managed if the safety of the individuals is determined before the feeding process to attain feeding efficiency. Food with different texture modification sizes and liquids with suitable consistency using thickener can be given adequate calories according to nutrient requirements. Food texture modification and liquid consistency might change according to an individual’s safety level and nutrient needs [37,40]. The International Dysphagia Diet Standardisation Initiative (IDDSI) framework was seen to help in providing a common terminology to explain food textures and liquid thickness [38].

Some interventions were positioning during feeding, correct food textures, and liquid consistencies. In addition, modification of the sensory properties of the food, oral motor facilitation, and adaptation with specialized utensils was also among the treatment for swallowing difficulties [37].

When identifying the challenges of feeding problems among individuals with CP, questions should be addressed on who, what, when, where, and how. “Who” is the person involved in the feeding process? “What” is the type, texture, viscosity, quantity, and quality of the food consumed? “When” is when the feeding process takes time, the frequency and duration of the meals? “Where” is where the feeding takes place and if any distraction was noticed, and “How” is how the feeding process occurred, such as the routine, technique, positioning, and adaptive equipment that assist during the feeding process? Note that the proposed questions to consider regarding the feeding problem were supported by another review study [46]. Three factors can affect the nutrition and growth of CP individuals, which are nutritional and non–nutritional factors. Nutritional factors include the feeding problem, in which the intervention mainly focuses on modifying their food and liquid consistency in the feeding process, decreasing calorie loss, and increasing energy needs for non-nutritional factors such as age, genetic, physical, neurotrophic, and mechanical stress on bones [39].

## 4. Discussion

The purpose of this scoping review is to discover the challenges of individuals with cerebral palsy (CP) and their caregivers when preparing modified textured food and to explore the components of nutrition management with an emphasis on food texture modification.

Based on the findings, the objective of the qualitative study was to find out the difficulties of individuals with CP who experienced feeding problems. Challenges reported faced by individuals with CP were the difficulties with swallowing influenced by food textures that might give the risk of choking. Other than the feeding process became more complicated when the environment was not friendly to them. Consequently, they tend to depend on social support and have negative feelings such as shame and frustration while trying to adapt to food [33]. The feeding problem affects older individuals with CP and caregivers with children with CP [29]. The cross-sectional study from the finding aims to determine the incidence of oropharyngeal dysphagia (OPD) with CP and its correlation with feeding duration, frequency, and feeding ability and to explore the growth, nutrition intake, and feeding behaviors of children with CP who could not consume hard food textures. In addition, a high Gross Motor Function Classification System (GMFCS) was related to feeding problems as they took a long time to finish food [30,32,36]. Subsequently, caregivers also felt overwhelmed when preparing food for their children with CP according to safety level as their children are prone to experience undernutrition in contrast to typical children [31,35,36]. Other factors that might contribute to feeding problems were proper positioning while feeding, considering taste and amount of food given, longer feeding time, and tackling sensory-based problems among individuals with CP [30,41].

Safety, stress level and nutritional status were among the most featured factors and challenges of food texture modification among individuals with CP and caregivers [15,18,32,33,35,36]. Thus, the following write-up will further discuss the highlighted factors/challenges experienced and recommend food texture modification as a nutrition intervention.

### 4.1. Challenges of Feeding Problems among Individuals with CP

During the process of swallowing, oral motor function, swallowing and respiration must be well coordinated. For individuals that experienced neurological impairments, particularly oral motor dysfunction, the swallowing process may not operate accordingly and lead to the risk of aspiration. Additionally, prolonged aspiration may lead to acute pneumonia, risk of infection, and eventually chronic lung damage [47]. Incidence of aspiration happens when food and fluid enter the airways and penetrate the lungs. Silent aspiration might also occur when no sign of coughing or respiratory problems emerges, but the food and liquid particles have already entered the lungs [48]. Feeding problems can lead to significant complications later in adulthood if not tackled in early life. In a population-based study among adults with CP (aged 37–58 years old), dysphagia was the third highest health condition reported (29%), with the most common health problem being due to pain (65%) and followed by upper gastrointestinal disorders (33%) [49]. Other complications associated with feeding problems were oropharyngeal dysphagia, aspiration pneumonia, intestinal constipation, and gastroesophageal reflux disease (GERD) [8].

In a study by Remjin et al. [33], safety was among the crucial aspects of eating and drinking in adults with CP. Participants valued eating and drinking as an occasion with different impacts and experiences. Consequently, the participants also tended to avoid food when triggered by coughing, as it was one of the indications that could lead to choking. Some of them would rather avoid the food that brings difficulties to them, leading others to strategize the feeding sequence and practice feeding from time to time [33]. A small-scale study performed among 37 children with CP aged 11–58 months, 32 children that were fed orally, found that the frequency of coughing was 56% and choking was 66% [50], and this is further demonstrated in another study in Children Hospital Lahore, those feeding problems involved drooling (66.7%), choking, and vomiting while eating (30%) [51].

Every healthcare profession has specific responsibilities and aims to treat individuals with CP feeding problems. For example, physicians must treat medical issues such as GERD, cleft palate, etc., that may interfere with feeding activity. The feeding specialist performs an instrumental evaluation to ensure safe feeding. Simultaneously, dietitians help supply sufficient nutritional intake and hydration according to the individual requirement. Psychologists may help with the individual’s behavioral issues with CP pertaining to feeding problems. Note that individuals with CP may consult social workers if the family’s economic status is affected [37]. This was supported by a review from Sharp et al. [52], which emphasized that thorough and multidisciplinary healthcare professionals’ approach advantages for children with feeding problems.

The choice of procedure, such as clinical bedside examination (CBE) and videofluoroscopy (VFSS), depends on the patient’s condition. In Malaysia, the determination for patients to undergo the VFSS procedure is based on the physician’s clinical assessment. It is conducted in the radiology department with supervision and guidance from speech therapists and physicians. The patient will be given different types of food texture modification, further determining the suitable rehabilitation treatment that can prevent aspiration [53]. Correspondingly, the procedure to confirm safety in swallowing involves testing various types of food texture modification, ranging from puree, and soft texture, to texture that is close to normal depending on the tolerance and safety of the procedure. Based on the findings, healthcare professionals will later advise and prioritize the suitable level of food texture that is safe for caregivers to prepare at home.

Feeding problems could lead to a deficiency of certain nutrients. Individuals with CP might experience not eating and drinking enough due to feelings of fear and worries that can lead to undernutrition. Moreover, worry arises as they seek healthcare treatment and do not discuss further nutritional issues [33]. The research done by Caramico-Faber et al. [54] found that for symptoms related to gastrointestinal problems, the prevalence of dysphagia was 33 subjects out of a total of 40 among children with CP (82.5%), and the average energy intake was low for those having swallowing problems as compared with none in children with CP. It was supported by a study done by Rajikan et al. [55] to determine growth rate, nutritional problems, and nutritional status among CP children found that 29.5% of them were underweight and had low triceps fold skin readings by age (skinfold-weight-age) (36.2%) (n = 153). A systematic study performed by Speyer et al. [6] in 42 articles reported the prevalence of swallowing problems was 50.4% and 53.3% for nutritional issues resulting in a higher risk of malnutrition among CP patients. This is also supported by another study conducted in a hospital in Surakarta, Indonesia, which found that 78% had difficulty eating, and the prevalence of malnutrition was 68% [56].

Dysphagia in CP children and the need for food texture modification is correlated with lower weight for age, a lower height for age, and lower calorie consumption in contrast to typical children [35]. This is due to the decreased energy-dense food than the typical texture. Other factors that might have caused reduced calorie intake were calorie loss due to choking or vomiting while eating, oral motor dysfunction, and caregivers’ failure to prepare correct consistencies [35]. As a summary of the data from the previous literature review, it is found that the constant findings of cases of malnutrition and nutritional problems among the population of CP children

### 4.2. Challenges of Feeding Problems among Families with CP

Stress can happen regardless in individuals with CP or among caregivers. Individuals with CP describe stress in a social place due to dependency on feeding utensils, accessibility, setting environment prior to eating, and dependence on caregivers. They also felt stressed by the limited menu choice, as they needed to choose texture over taste for safety purposes. Note that longer mealtimes caused frustration as they needed to limit their social activities and felt shame when their surroundings gave an odd look while they struggled to use adaptive utensils while eating [33]. Fatigue while eating can be one of the reasons for prolonged mealtime [13].

The caregiver’s stress can be defined as a parental burden [57]. The daily activities, including feeding, can be challenging to the caregivers. Some qualitative studies exploring caregiver experience of feeding difficulties among children with CP found that the caregivers’ stress was influenced by worrying if their children were choking while eating and not getting enough calories [29]. Other than that, several principal caregivers revealed that CP children were given food by mouth and the remaining fed via gastrostomy because of this concern [58]. Caregivers who fed orally found the feeding process to be cherished. However, they found it physically and emotionally stressful, causing pain in the neck, shoulder, and back to maintain the position while giving food, and more challenged if their children have muscle stiffness [58].

Issues raised among children with CP guardians were the need to prepare food according to the texture of food that is safe (particularly food with a texture of solid, big size, lumpy, hard, and tough), taking longer time to prepare the food and child’s behavior during mealtime [33,35]. High-level stress is also correlated with the severity of CP and increased feeding difficulties. This was also agreed by a study done by Polack et al. [59] among 76 caregivers in Ghana showed that eating problems were closely related to the quality-of-life scores of caretakers. The eating process will take longer in food preparation, resulting in a lack of time for caregivers to generate more income to improve their socioeconomic status. This study shows that caregivers were more concerned with the emotions experienced during the feeding process compared to concerns about the child’s lack of nutrition and weight. More in-depth studies through qualitative methods need to be conducted to understand the issue of nutritional problems with the well-being of caregivers [60].

The behavior of individuals with CP and caregiving demand influences caregivers’ psychological and physical health, which leads to stress [16,61,62]. Therefore, strategies to overcome feeding problems might include behavioral and stress management [61].

### 4.3. Nutrition Management with Food Texture Modification

Food texture modification was the treatment for feeding difficulties among individuals with CP. In research conducted by Benfer and colleagues [18] on 99 infants and toddlers with CP (aged between 18 and 36 months), 39% consumed modified textures of food. Modifying food texture was one of the essential components of the training program besides introducing calorie-dense food, appropriate utensils to use, positioning, and feeding practices [63].

A multi-professional group developed the International Dysphagia Diet Standardization Initiative (IDDSI) framework in 2013 to assist dysphagia individuals with standard techniques in food preparation according to swallowing safety based on the degree of modification of food texture and fluid consistency [64]. Studies show that the implementation of IDDSI can resolve the issue of inconsistent use of terminology for various types of food texture modification. Other than that, a study conducted in a hospital in Germany through a pilot campaign showed that the implementation of IDDSI requires multidisciplinary cooperation, such as medical officers, nurses, speech therapists, food service divisions, and dietitians [65]. Joint meetings were held, and patients requiring modification of food texture and fluid consistency were identified. Specific terminology is used for communication with the food service department. The head chef is responsible for developing a new menu according to the suitability of the food texture modification, and digital software is used as a medium to key in information on IDDSI along with descriptions to facilitate the process of food preparation. Consequently, the dietitian is assigned to provide exposure regarding IDDSI and adapt it to the organization. Implementing the IDDSI framework is helpful in terms of standard terms for the diversity of food textures and as a space for communication with multidisciplinary health professionals [65]. Moreover, the latest research found that the IDDSI implementation helps the aged care population by improving patient compliance, consuming a texture-modified diet, and increasing staff knowledge of the terminology recommended by the IDDSI framework [66]. To date, IDDSI implementation is not yet done in individuals with special needs, especially those with CP. Caregivers are the closest people to individuals with CP. Therefore, preparing food texture modification according to the needs and safety level should be easy even with standard guidelines. A previous study conducted among 59 university students in preparing food textures according to IDDSI guidelines agreed that following the guidelines was feasible and straightforward. It was observed that 90% of them acknowledged that food preparation was not time-consuming [17]. A study on the low-income population found that children with CP somewhat suffered from undernutrition; 86% in Bangladesh, 98% in Indonesia, 72% in Nepal, and Ghana. Note that maternal education was also found to link with the undernutrition status among these children with lower functional disabilities associated with feeding problems [67]. Maternal education influences the knowledge of health-related factors which govern decision-making in seeking treatment and enhancing the nutritional status of their children [68].

There is a promising possibility in the field of food texture modification. Currently, there are many efforts made by various disciplines to help individuals who experience swallowing difficulties that require food texture modification. Besides families that prepare food texture modification at home, healthcare settings and the food industry were seen as a potential part of preparing food texture modification. Additionally, a new technology of three-dimensional (3D) food printing, as mentioned by Lorenz et al. [69], may help the food industry by designing food mold according to real food by forming food mold layer by layer. This technology can help reduce human resources in preparing food texture modification. However, in a qualitative study among healthcare professionals (n = 15), the finding showed that 3D food printing is costly and a lengthy process in handling the equipment [70]. In the current practice in the hospital setting, the textured-modified food preparation is done by the food service department, mainly using a thickening agent to adjust fluid consistency according to the patient’s needs and safety. Hence, the 3D technology of food texture modification is exciting and worth going through the development process to further benefit individuals with swallowing problems and their families.

## 5. Conclusions

This study has highlighted factors and challenges in food texture modifications among individuals with CP. The foreground is related to feeding-related issues, including gastrointestinal problems, choking, aspiration, tube feeding, and food texture modification. Nevertheless, the types of food texture modification, such as the terminology with characteristics of the food texture and preparation techniques such as cutting method, cooking style or testing method to get the desired level of food texture that is safe for individuals with CP experiencing swallowing problems were not explained in detail. Currently, there are no standard guidelines, and with varied practices, parents find it difficult to sustain or progress nutritionally with their children’s needs. The consequences are parents might follow common nutritional advice and continue practicing recommendations suggested several years back, as there was no prioritization on following up for dietary intake. As the safety level is confirmed, preparing according to the food texture modification can improve caregivers’ quality of life and their children’s nutritional status. Note that food texture modification involves clinical examination by healthcare professionals for food preparation at home. It is acknowledged that multiple approaches are essential in addressing this issue. Therefore, primarily offering solutions relating to food preparation based on a safe texture level is crucial and achievable for caregivers in family institutions.

### Limitations and Future Studies

The study within the scope of this area is limited, and the database also restricts the findings discussed in this paper. Some research encompasses individuals with CP and a cluster of neurological disorders, which limits the search manner of this review. Furthermore, empirical data on this area is deficient. Since CP is a diverse group, a qualitative study should be further encouraged to explore the challenges faced by this group in dealing with food texture modification besides using validated questionnaires to assess the quality of life of feeding problems among caregivers. Authors should discuss the results and how they can be interpreted from the perspective of previous studies and of the working hypotheses. The findings and their implications should be discussed in the broadest context possible. Future research directions may also be highlighted.

## Figures and Tables

**Figure 1 nutrients-14-05241-f001:**
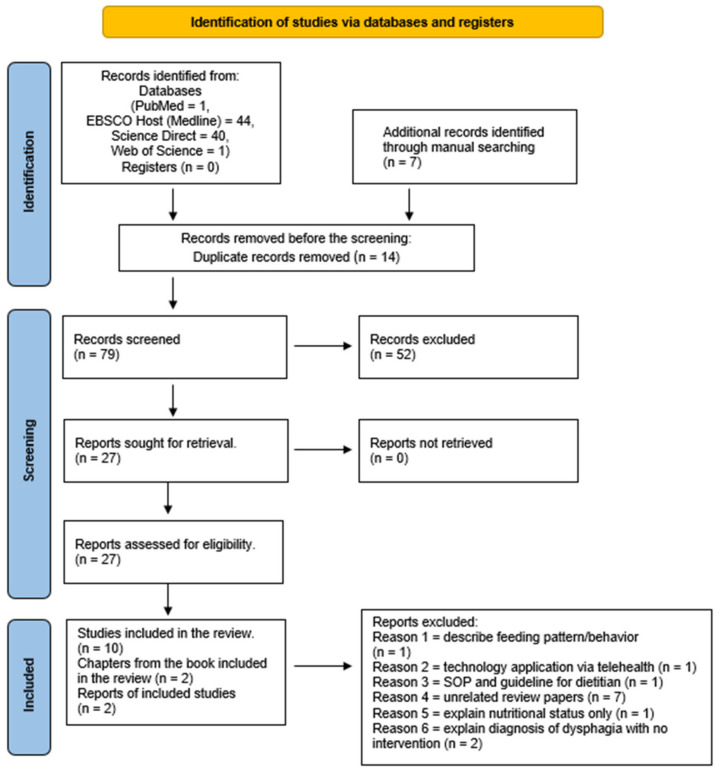
PRISMA-SCR 2020 Flow Diagram.

**Table 1 nutrients-14-05241-t001:** The Population, Concept, and Context (PCC) Table.

Patient or Population	Cerebral Palsy
Concept	Food texture modification
Context	swallowing difficulties (dysphagia)

**Table 2 nutrients-14-05241-t002:** List of keywords and synonyms generated as search terms.

Keywords	Synonyms	Variation
Texture	Consistency	Textures, textured
	Composition	
	Constitution	
Modification	Adjustment	Modify, modification
	Adaptation	
Difficulty	Complication	Difficulties, complications, obstacles, struggles
	Obstacle	
	Struggle	
Disorders	Complications	Disorder, complication, complicated
Swallowing	Deglutition	
Challenges	Demand	
	Objection	
	Threat	

**Table 3 nutrients-14-05241-t003:** Challenges faced by individuals and families with cerebral palsy.

Author and Year	Location	Study Design	Sample Size and Age	Method	Findings/Key Points	Remark
Taylor (2022) [29]	Australia	Qualitative-semi-structured interview	Caregivers with CP aged less than 18 years (n = 11)	In-depth interviews by telephone	Child-centered world–worry due to choking and concern about safety and body weight, food preparation, food, and liquid modification, positioning, and longer feeding time. Making decisions–problem-solving depends on the child’s health status, seeking knowledge to treat feeding problems. Knowing their child–unique feeding feeds affect the food preparation process. Seeking and receiving support–family support, healthcare professionals’ intervention, financial issues, and social support.	Healthcare support is essential in managing feeding problems among individuals with CP.
Gonzalez (2022) [30]	Mexico	Case-control study	Eight months–15 years of age Control (CP with no OPD, n = 30) Case (CP with OPD, n = 30)	All patients undergo VFSS–testing with three consistencies: nectar, thin liquid, and puree consistency. GMFCS Weight, Height and BMI	OPD related to degree V GMFCS Longer feeding length above 30 min, lack of ability to keep the lips together, coughing during or after meals, choking all through or after meals The process of swallowing food in bolus form was significantly difficult (*p* < 0.05). It took a longer time in CP with OPD than with CP without OPD as perceived by the mother.	The Association between OPD documented by VFSS and the severity of gross motor function impairment ought to be due to a greater neurologic injury that led to feeding problems.
Garcia (2021) [31]	Spain	A descriptive, cross-sectional, open-label study	n= 44 (children and adolescents with CP)	GMFCS Dysphagia screening EDACS	Patients with more significant feeding difficulties (higher EDACS levels) presented more severe functional impairment (higher GMFCS levels) BMI values indicated poorer nutritional status in patients with higher EDACS levels and severe GMFCS.	Medical history interviews should include questions addressing the ability to bite, chew, and swallow and to manage different textures of foods and fluids, as well as red flags indicating dysphagia.
Mahmoud (2019) [32]	Egypt	Correlation-al study	Individuals with CP (n = 100) from 1–4 years.	SOMA GMFM	Eating capability was substantially related to gross motor functional abilities. Children’s capacity to consume food textures with advancing complexity was best in those with GMFCS I and progressively reduced as GMFCS level increased (or gross motor functional capacity decreased).	Future research investigating the occurrence of OPD using evaluation of feeding skills, relationship with capability on food textures, and effect on nutritional status throughout the full range of gross motor function capabilities are required.
Remijn (2019) [33]	Netherlands	Qualitative study	Participants with spastic CP (n = 10), aged 15–23 years	Semi-structured in-depth interviews	Perceived eating and drinking difficulties food textures influenced problems with swallowing and mastication Challenges in a physical and social context challenges in the accessibility as a wheelchair user, menu choices were determined primarily by texture over taste preferences, dependency on others Dealing with eating and drinking difficulties adaptation or avoidance of foods, perseverance to keep trying or giving up to try or acceptance of help Negative feeling feelings of shame, frustration, distress, and fear concerning choking	Healthcare providers and the food industry play a vital role in the CP population that needs food texture modification in their daily diet.
Yi (2019) [34]	Korea	A cross-sectional, interview-based survey study	Adults with CP on full oral diet (n = 117) Healthy individuals (n = 117)	SWAL-QOL GMFCS MACS FOIS	The majority of CP participants receiving complete oral diets had pharyngeal symptoms (choking on food, choking on liquid, coughing when food became stuck, coughing, clearing the throat, and food sticking in the throat) Most participants needed modification or restriction of certain foods due to longer meal durations and lower BMI.	In adults with CP, dysphagia symptoms are frequent and can profoundly affect swallowing related QOL.
Serel (2018) [35]	Turkey	Cross-sectional study	A study group with individuals with CP (n= 50 control group (n= 35), aged 18–90 months	24-h diet recall BPFAS GMFCS	Children with CP had a greater incidence of choking and vomiting during meals. Caregivers perceived the mealtime behavior of children with CP was greater problematic. Caregivers may fail to prepare correct food consistency. Lower energy intake in children with CP due to types of diet given, energy loss at some point of feeding. Types of diet taken were liquidized or pureed food. Children with CP whose diets consisted of food textures Levels 3 to 5 according to IDDSI classification had poorer growth and nutritional status. CP whose diet is fully composed of liquidized or pureed meals with no lumps may no longer meet their energy or nutrient due to decreased energy density and reduced calorie intake.	Parental reported the need for the preparation of appropriate food texture, preparation time constraints, and the child’s behavior during mealtime. The feeding issues of the inability to take chewable meals may contribute to the growth, dietary status, negative feeding behaviors, and greater problematic perceptions by caregivers among children with CP.
Marques (2016) [15]	Portugal	A mixed descriptive and exploratory study	CP with their families (n = 104)	Anthropometric Parents Questionnaire (Qualitative study) Family APGAR score	Parents stated that feeding issues were an unbalanced diet, meal monotony, shame, difficult access to tailored outside food, insufficient technical equipment, food spilling, choking, vomiting, time, cooking meals separately, and costs. Food aspiration is one of these risks, leading to severe problems such as pneumonia, airway obstruction, and even death. Parents regularly pick food that is quicker to cook, with equal consistency, which leads to no variety in food/nutrients.	Food consistency should be tailored to the child’s constraints and to minimize the risk of food aspiration.
Benfer (2015) [18]	Australia	A cross-sectional, population-based cohort study	Young children with CP (n = 99) aged 18 to 36 months	3-day weighed food records. PEDI DDS CFQ Swallowing Safety Recommendation	Modified food/fluid textures are common in children’s diets with CP. Modifications to diets limit their child’s ability on food textures, indicating parents are generally excluding foods/fluids for which they perceive their child has difficulty. Children with severe GMFCS consume a lower proportion of chewable foods and more fluids. Purees and fluids are likely more efficiently eaten by children with lower gross motor function.	Training parents to detect safety concerns on food/fluid textures with higher density may be more clinically meaningful and effective than focusing on identifying specific oromotor impairments to achieve adequate energy intake.
Benfer (2014) [36]	Australia	Cross-sectional study	Cerebral palsy (n = 130 Typical children (n = 40), aged 18–36 months	DDS SOMA PSAS Parent-report 3-day weighed food record GMFCS Feeding questionnaire	Oral phase impairments associated with GMFCS level. For solid food, children with CP had difficulty biting (70%), cleaning behaviors (70%), and chewing problems (65%).	Increasing energy density with the correct food texture improves nutritional outcomes.
**Author & Year**	**Location**	**Book Chapter & Report**		**Key Points**		**Remark**
Fleet (2022) [37]	USA	Encyclopedia of Human Nutrition (Third Edition), Academic Press, 2013, Pages 21–27 Pediatric feeding disorders: feeding children who can’t or won’t eat		Positioning, food texture, bolus size, rate, and the amount of food presented can determine the safest and most efficient feeding method.		The goal of safe oral feeding is attainable in most children when those involved in the care of children understand the complexity of eating and the associated medical and psychological conditions that comprise a feeding disorder.
Miller (2021) [38]	USA	Reference Module in Food Science Cerebral palsy: nutritional aspects		Changes in the texture of solids and liquids may be necessary to ensure safety. Management strategies for daily mealtime feeding include positioning, modification of the sensory properties of the food, oral motor facilitation techniques, and equipment adaptations. The International Dysphagia Diet Standardization Initiative (IDDSI) Framework provides a common terminology to describe food textures and liquid thickness.		All interdisciplinary plans, including nutrition, should be safe, promote growth or weight gain without excessive energy expenditure and reflect the family’s resources in time and skill, addressing their concerns and expectations.
Rempel (2015) [39]	Canada	Physical Medicine and Rehabilitation Clinics of North America Volume 26, Issue 1, Pages 39–56 Good Nutrition in Children with Cerebral Palsy		Valuable information in understanding a child’s feeding challenges WHO: Persons involved with feeding; differences in feeding styles WHAT: The type, texture, viscosity, quantity, and quality of the food consumed WHEN: The timing, frequency, and duration of meals WHERE: The feeding environment, distractions HOW: The feeding routine, technique, adaptive equipment, and positioning Nutritional factors that require food texture modification help in the development of children with CP.		Consideration of the multidimensional aspects of feeding and the contribution of family members in setting goals and carrying out the nutritional intervention.
Cohen (2011) [40]	USA	Pediatric Gastrointestinal and Liver Disease (Fourth Edition) Pages 1020–1032.e3 Chapter 92—Nutrition and Feeding for Children with Developmental Disabilities		Taste and texture can be different if the individual is identified as safe to eat orally. Decreased feeding efficiency happens in individuals with CP: chewing and swallowing take 12 to 15 times longer than in typical individuals, and lead to insufficient caloric and nutritional needs. Individuals with sensory-based problems had greater issues with foods that require chewing and may separate foods of thicker textures and pocket them in their mouths. In addition, they frequently have a sensory integration defect with the texture and taste of foods.		Caregivers must be adequately instructed, trained, and reassured, and appropriate follow-up must be arranged to verify the patient’s progress and alter the routine to obtain an optimal outcome.

BPFAS = Behavioral Pediatrics Feeding Assessment Scale. CP = Cerebral Palsy. CFQ = Child Feeding Questionnaire. DDS = Dysphagia Disorder Survey. EDACS = Eating and Drinking Ability Classification System. FOIS = Functional Oral Intake Scale. GMFCS = Gross Motor Function Classification System. GMFM = Gross Motor Functional Measure Scale. MACS = Manual Ability Classification System. OPD = Oropharyngeal Dysphagia. PEDI = Pediatric Evaluation of Disability Inventory. PSAS = Pre-Speech Assessment Scale. SOMA = Schedule for Oral Motor Assessment. SWAL-QoL = Swallowing-quality of Life. OPD = Oropharyngeal Dysphagia.

## Data Availability

This manuscript does not contain original data.

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
