# Peer review of "A Review of Food Texture Modification among Individuals with Cerebral Palsy: The Challenges among Cerebral Palsy Families"

_nutrients, 2022, doi:10.3390/nu14245241_

Round 1
Reviewer 1 Report
The scoping review aims are relevant and useful to the community. As such the work has potential but there are substantial revisions required.
The abstract is clear, however, the final conclusion that Standard guidelines could be developed, is not sufficiently supported by the preceeding information. Are there already standard guidelines that are not being used or that are not suitable?
Paragraph starting line 67 needs revising. Specifically, why is Erasmus et al chosen as the guidance for assessment? And Videofluoroscopy is the instrumental procedure of choice for children with CP, although in some instances FEES might be preferable.
Line 78 - "treatments are necessary" - It is for parents to decide what is necessary. And the treatment/ intervention/ support might improve safety and/ or enjoyment and/ or nutritional status.
Line 81 "The aim of feeding must be practical..." could be "families need to be involved in determining goals and activities in a feeding plan, to ensure the plan can be effective for the family".
Can you expand on what is "oral sensori motor" as this is not clear - are you referring to oro-motor exercises?
NMES is still very controversial.
Line 115 - "Mentally handicapped children" is not current terminology.
It seems important to include IDDSI in the introduction. It is not mentioned at all explicitly. This would be valuable in complying with standardisation of terminology for textures.
The article: Assessment of International Dysphagia Guidelines for Use in Child Nutrition Programs - Shaw, Catherine; Moore, McKenzie; Camel, Simone; Douglas, Crystal Clark - Journal of Child Nutrition & Management, v43 n2 Fall 2019. Seems a relevant example of a training programme that directly uses IDDSI and has methods for testing distinct food textures.
Table 3 - OPD is used but not spelt out in full - an asterix would help guide the reader to the table notes.
Table 3 - this does not make sense "greater difficulty in food bolus transportation pronounced perceived by the mother among children with CP with OPD."
Some subheadings or structure to the results would help readability and relate back to the aims of the study. The focus on relating severity of problems to GMFCS does not seem particularly relevant to the aims of the study and the aims should be clarified. The emphasis in the aims is on 'preparing' modified texture food.
Line 83 of results - "swallowing longer" This is not clear and should be rephrased.
The section on safety again does not seem directly relevant to the aims. Paragraph starting line 176 about assessment does not appear directly relevant to the aims of the study. More direct linkages to modified food textures is needed throughout the discussion.
It is not clear how the study meets the aim to: Prioritise the provision of different types of food texture modification.
Reviewer 2 Report
The article is interesting and well-organised. I have some doubt about using term "food texture" which is well described in literature. In article Authors used it in the same meaning as consistency and form of food (liquid, puree, chopped food).
In the conclusions Authors reported that line 311-312
“However, the types of food texture modification and preparation
techniques were not explained in detail”
I am surprised because in literature it many examples of using different techniques to modify the food products by adding the gelling agent for consumers with dysphagia. Also now, there some examples of studies concerning application of 3D printing for modify food consistency.
In my opinion this apart should be added (the article about the possibilities of food modification)
Round 2
Reviewer 1 Report
A considerable effort has been made to address my previous points. However, the manuscript still does not meet the intended aims and I recommend a substantial re-write, rather than just changing some sentences.
The title "...what makes it count" does not make sense and is not answered obviously in the manuscript. I suggest that part is removed.
The aims of the study are not clear - is the study about feeding problems for individuals with CP, or is it about food texture modification? I think that throughout the manuscript the word "prioritization" is not accurately used and is leading to confusion. Possibly, a more accurate word would be 'recommendation'.
The results section should be re-written in direct response to the two research questions asked. The questions could be used as subheadings. This would make the results more clearly related to the aims of the study.
Regarding the previous points:
1. Abstract, the final sentence still recommends guidelines and still requires requires clarification as the rest of the abstract does not mention guidelines. Perhaps the manuscript needs to be clearer if no guidelines were found and that is something people have identified as missing.
2. There is still minimal justification for the use of Erasmus et al for clinical assessment, however, a multidisciplinary approach is good practice. The final sentence of this new paragraph requires rewording..."It is recommended that children with possible dysphagia are examined using an instrumental procedure like VFSS using barium, to gather information about the pharyngeal phase of the swallow and potential silent aspiration."
3. amendment is fine now.
4. amendment is fine now.
5. This requires a reference as oro-motor exercises are controversial and need to be carefully chosen.
6. good.
7. good.
8. Good.
9. These new additions are good and should also be referred to in the discussion as they relate to standard guidelines.
10. good
11. good.
12. good.
13. I would still recommend changing this wording, as it is not only the per-swallow time that is longer, but also the meal overall.
14. this is an improvement but the wording still requires modification in order to make sense. The section on safety is focused on assessing safety, yet the aim of the study is to "identify challenges of individuals with CP and their caregivers when preparing modified textured food and to explore the components of nutrition management and prioritize the provision of different types of food texture modification". The discussion section on safety should be rewritten, to focus on how texture modification claims to be safer, whether professionals are recommending texture modification, whether families perceive it as safer, whether families accurately prepare the food so it is safer and other issues that are arising in the scoping review about the safety of texture modification. There are many aspects of safety in relation to food texture modification that would be of more benefit here than a discussion about assessment.
15. as with 14., this is an improvement but the wording requires modification. Additionally, whilst safety should be a high priority, quality of life is potentially equally important and this might be something that has not emerged from the literature review but should be considered as a reason why families are not following guidance.
Author Response
Dear Reviewer,
Thank you for the comments and suggestions; we have thoroughly reviewed the manuscript and did restructure the manuscripts according to research objectives. The manuscript also had been proofread by a certified proofreader. Please see the attachment for the responses.
